

# Drought propagation in high-latitude catchments: Insights from a 60-Year Analysis Using Standardized Indices

Claudia Teutschbein[1], Thomas Grabs[1], Markus Giese[2], Andrijana Todorović[3], and Roland Barthel[2]

[1] Uppsala University, Department of Earth Sciences, Program for Air, Water and Landscape Sciences, Villavägen 16, 75236 Uppsala, Sweden
[2] University of Gothenburg, Department of Earth Sciences, Medicinaregatan 7B, 41262 Gothenburg, Sweden
[3] University of Belgrade, Faculty of Civil Engineering, Institute of Hydraulic and Environmental Engineering, Bulevar kralja Aleksandra 73, 11000 Belgrade, Republic of Serbia

*Correspondence to*: Claudia Teutschbein (claudia.teutschbein@geo.uu.se)

**Abstract.** Droughts, traditionally less associated with high-latitude regions, are emerging as significant challenges due to changing climatic conditions. Recent severe droughts in Europe have exposed the vulnerability of these northern catchments, where shifts in temperature and precipitation patterns may intensify drought impacts. This study investigates the dynamics of drought propagation in high-latitude regions, focusing on four key aspects: (1) the typical lag time for drought conditions to

propagate from initial precipitation deficits to impacts on soil moisture, streamflow, and groundwater systems, (2) the probability of precipitation deficits leading to these droughts, (3) the key factors influencing drought propagation, and (4) how drought propagation has evolved under changing climate conditions. By analyzing long-term observational records from 50 Swedish catchments, the study reveals that drought propagation is highly variable and influenced by a complex interplay of catchment characteristics, hydroclimatic conditions, and soil properties. Soil moisture exhibits the shortest propagation times,

often responding within a month to precipitation deficits, while groundwater shows the longest and most variable response times, sometimes exceeding several months. The probability of precipitation deficits propagating into soil moisture droughts is highest, followed by streamflow and groundwater, with these probabilities increasing over time. Across all drought types, annual precipitation and streamflow are the strongest governing factors, driving both propagation time and probability. Despite ongoing changing climate, drought propagation times or probabilities have not significantly changed over the past 60 years.

However, while most catchments are becoming wetter across all seasons, southern catchments become more vulnerable to spring drought due to increased evaporative demand. These findings highlight the need for tailored, region-specific water management strategies to address seasonal and regional variations in drought risks, particularly as climate change continues to evolve.





**Non-technical summary:** Droughts are becoming a growing concern in high-latitude regions, like Sweden, due to changing climate patterns. This study looked at how droughts develop and spread through soil, rivers, and groundwater systems by analyzing data from 50 catchments across Sweden. Overall, most areas are getting wetter throughout the year, but the southern regions become more prone to spring droughts due to increased evaporation.

The study found that soil moisture responds quickly to a lack of rain, within about a month, while rivers take around two
months to show signs of drought, and groundwater takes the longest to respond - up to several months. The chance of a rainfall shortage leading to a drought is highest for soil moisture, followed by rivers and then groundwater. The biggest factors influencing droughts are the amount of rain a region gets and how much water flows through rivers.

Importantly, despite changes in climate, the study found that the underlying drought dynamics stayed stable over the past 60 years. This means that current water management strategies based on past data are still reliable, but ongoing monitoring will
be essential as climate change continues to evolve. The findings highlight the need for region-specific approaches to managing water resources, especially in southern areas where spring droughts are becoming more of a concern.



# 1 Introduction

Droughts, typically described as episodic, socio-climatologically induced water deficits caused by anomalies in average
conditions (Pereira et al., 2006), can occur in any climate zone (Wilhite, 1996; WMO and GWP, 2016). They stand out among
natural hazards due to their unique characteristics, progressing gradually with a slow onset and long recovery times, making
them difficult to identify precisely (Rajsekhar et al., 2015; Spinoni et al., 2015; UNDRR, 2021). Often referred to as 'creeping
disasters' (Van Loon, 2015), they are subtle but can persist for extended periods and have considerable impacts across various
sectors, including water supply quality and quantity, ecosystems, agriculture, and hydropower production (Blauhut et al., 2022;
Teutschbein et al., 2023b; UNDRR, 2021).

Although droughts are typically associated with arid or semi-arid regions, recent events such as the 2018 or the 2022 European
droughts (Blauhut et al., 2022; Garrido-Perez et al., 2024; Tripathy and Mishra, 2023) have shown that colder high-latitude
regions, including Scandinavia, are also at risk (Bakke et al., 2020; Teutschbein et al., 2022, 2023a). High-latitude catchments,
which feature distinct hydrological processes such as snow accumulation or snowmelt, exhibit unique responses to climate
change and drought conditions. These regions are particularly sensitive to climate change (IPCC, 2021) and have already seen
substantial alterations in streamflow regimes (Arheimer and Lindström, 2015; Blöschl et al., 2017; Teutschbein et al., 2022). As the climate further warms, changes in temperature and changing
precipitation patterns are expected to disrupt snow-related processes even more, leading to profound shifts in the hydrological dynamics in these regions (Arheimer and Lindström, 2015; Irannezhad et al., 2015; Skålevåg and Vormoor, 2021; Teutschbein et
al., 2015; Wilson et al., 2010). These changes are likely to intensify the development and propagation of droughts (Ahopelto et al., 2023; Spinoni et al., 2018; Teutschbein et al., 2023b), with complex and multifaceted hydrological consequences.

This complexity arises because drought development involves changes in water fluxes that affect various feedback mechanisms within the hydrologic cycle (Van Loon, 2015). Drought propagation is generally understood as a hierarchical
top-down process (Changnon, 1987), where

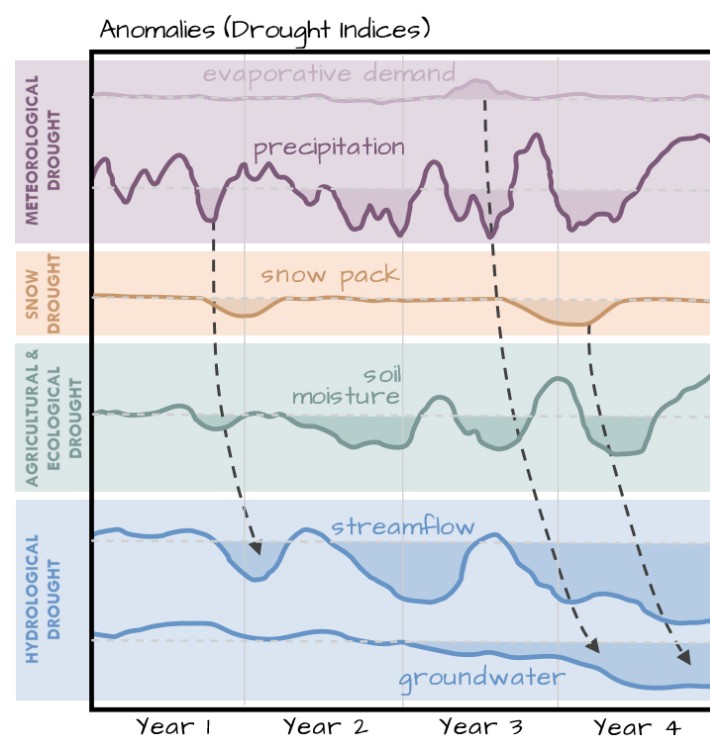

**Figure 1: Conceptual framework depicting the onset and hierarchical progression of drought within the water cycle (adopted and modified from Changnon, 1987). The framework begins with meteorological drought and illustrates the cascading effects on snowpack and soil moisture, as well as on streamflow and groundwater.**



variations in precipitation (rain/snow) and temperature (as proxy for evaporative demand) cause a meteorological drought (Figure 1). Over time, this can cascade down to other hydrological variables in the water cycle, e.g., to snow pack, soil moisture, streamflow, and groundwater, often occurring in a non-linear manner and with considerable delays (Mukherjee et al., 2018). Deficits in soil moisture that affect soil vegetation and crops are typically framed as agricultural/ecological droughts (Van

Loon, 2015), and deficits in streamflow and groundwater as hydrological droughts (Mishra and Singh, 2010). These various types of drought are interlinked through both positive and negative feedback processes (Van Loon, 2015), and the seasonal dynamics of snow and ice add further complexity to the processes. The timing and intensity of snow accumulation and melt can significantly influence water availability, either counteracting or amplifying seasonal precipitation deficits (Staudinger et al., 2014).

While drought propagation has been examined in other parts of the world, particularly in more temperate and arid regions (Bevacqua et al., 2021; Brunner and Chartier-Rescan, 2024; Entekhabi, 2023; 2024a; Heudorfer and Stahl, 2016; Odongo et al., 2023; Sattar et al., 2019), and with detailed analyses often focusing on individual events (often referred to as 'storylines'; see, e.g., Chan et al. (2022) or Gessner et al. (2022)), high-latitude catchments have not received the same level of systematic and statistical attention. In particular, there is a significant gap in research that comprehensively accounts for the heterogeneity

in catchment sizes, topographic features, land-use patterns, and climatic conditions unique to these regions. This gap hampers our ability to discern common patterns and key factors influencing drought propagation, resulting in an incomplete understanding of how droughts evolve and transition across these landscapes.  To address these knowledge gaps, this paper seeks to answer the following research questions in the context of high-latitude catchments:

1.   What is the typical lag time for drought conditions to propagate from initial precipitation deficits to subsequent
95        impacts on soil moisture, surface water, and groundwater systems in high-latitude catchments?

2.   What is the propagation probability of precipitation deficits translating into droughts in soil moisture, surface water, and groundwater in these regions?

3.   How has drought propagation in high-latitude catchments evolved in response to a changing climate?

4.   What are the key factors influencing the propagation and progression of droughts in high-latitude catchments?

## 2  Methodology

### 2.1  Study Sites

The propagation of drought was analyzed using the CAMELS-SE dataset (Teutschbein, 2024a, b), which includes data from 50 high-latitude catchments in Sweden (Figure 2), spanning the years 1961 to 2020. These catchments are distributed across a longitudinal range from 56°N to 68°N and encompass all three major climate zones in Sweden (Figure 2a): the polar tundra

climate (ET) in the Scandinavian Mountains of northwestern Sweden, the subarctic boreal climate (Dfc) in central and northern



Sweden, and the warm-summer hemiboreal climate zone (Dfb) in southern Sweden (Teutschbein, 2024a; Todorović et al., 2024).

Average elevation across the catchments varies from 12 to 942 m a.s.l. (Figure 2b), with catchment areas ranging from 2 to 8,425 km². Forests dominate the land cover of these catchments, with only very few catchments exhibiting intensive

agricultural activities.  Glaciers and urbanized areas occupy negligible portions of the catchment area, up to 2% and 3% respectively, while lakes and wetlands are generally scarce, with a median area of 12%. Approximately one-third of the catchments are subject to regulation, though the impact of reservoirs on streamflow is relatively minor (Todorović et al., 2022; Tootoonchi et al., 2023), which is crucial for accurately representing natural drought propagation.

The catchments exhibit different hydroclimatic properties, and can – following the grouping by Teutschbein et al. (2022) - be

clustered in five different groups based on their streamflow dynamics (Figure 2c). The selected catchments are predominantly humid, with the wettest areas found in western Sweden, characterized by both high precipitation and streamflow (clusters 1, 3 and 4). Snow-dominated and transitional catchments (clusters 1-3) are more common than those dominated by rainfall (clusters 4-5). A noticeable north-south temperature gradient exists, with catchments in clusters 1 and 2 featuring mean temperatures below zero, catchments in cluster 3 around 1.6°C and in clusters 4 and 5 around 6.5°C (Figure 2c). A detailed description of

the physiographic and hydroclimatic features of the catchments can be found in Teutschbein (2024a, b)

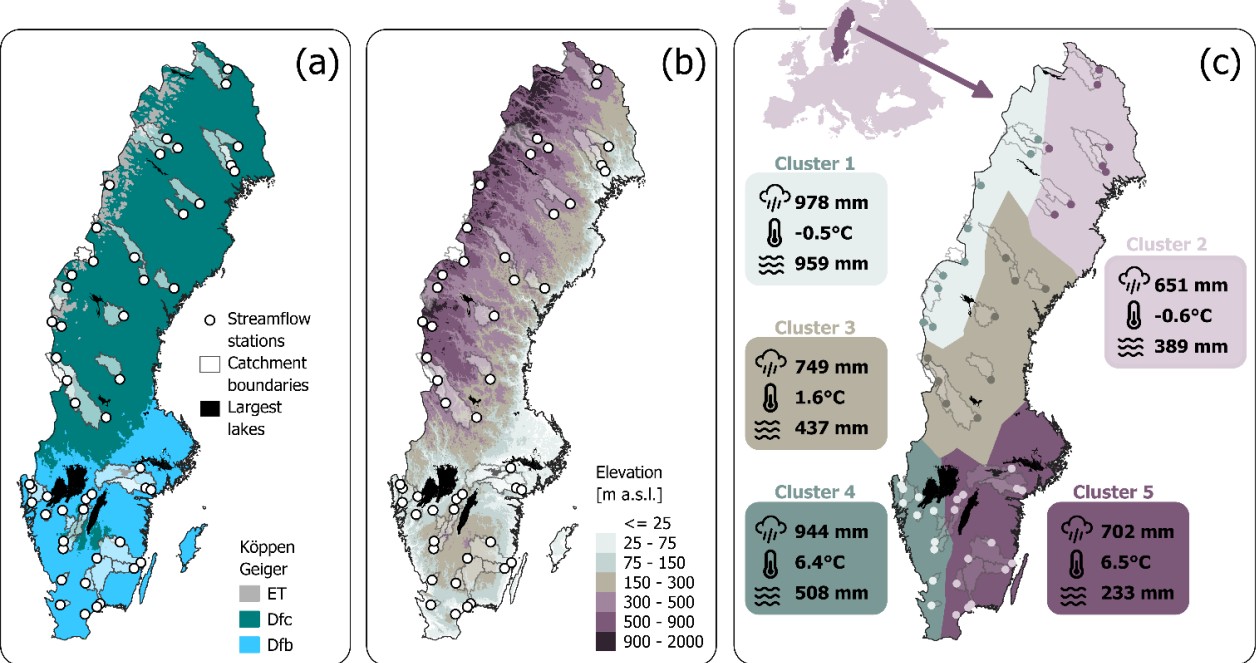

**Figure 2: Illustration of the 50 streamflow stations along with their catchment areas in relation to Sweden's (a) climate zones, as classified by the Köppen-Geiger system (Beck et al., 2018), encompassing the polar tundra (ET), subarctic boreal (Dfc), and warm-summer hemiboreal (Dfb) climates, (b) elevation data sourced from Lantmäteriet, the Swedish Mapping, Cadastral, and Land**

**Registration Authority, and (c) hydroclimatic catchment clusters identified by Teutschbein et al. (2022) with their corresponding annual precipitation, mean tempereature and streamflow statistics.**





## 2.2 Data

Daily precipitation, temperature and streamflow data from 1961 to 2020 were obtained from the freely available CAMELS-SE dataset (Teutschbein, 2024a, 2024b) and aggregated into monthly mean values. The daily temperature and precipitation
series in the dataset were originally derived from the Swedish Meteorological and Hydrological Institute (SMHI) national precipitation-temperature grid, which has a spatial resolution of 4 km x 4 km (SMHI, 2005; Johansson, 2000).

Monthly mean volumetric soil moisture for the uppermost 7 cm of soil over the same period (1961-2020) was obtained from the ERA5-Land reanalysis dataset, which is publicly available at a spatial resolution of 11 km x 11 km through the Copernicus Climate Data Store (Hersbach, et al., 2023). Catchment-specific values for precipitation, and soil moisture were computed by
averaging all grid cells whose centers were fully located within the catchment boundaries.

Monthly mean groundwater observations were downloaded from the Swedish Geological Survey (SGU), which provides measurements across 1,506 wells across Sweden on its publicly accessible website. For this study, we selected wells based on the following criteria:

- located within the boundaries of any of the 50 study catchments or within a 5km buffer
- minimum of 20 years of data coverage
- at least 120 data points
- no gaps in the data lasting more than 5 months

A spatial match between groundwater wells meeting these criteria and catchments was found for only 15 out of the 50 catchments, enabling the study of drought propagation throughout the entire hydrological system in these areas. The number
of groundwater wells within the 15 catchments varied from 1 to 14. Data gaps up to 5 months were filled using the average of normalized groundwater levels from the remaining wells in the catchment (when more than one well was available), otherwise through linear interpolation.

## 2.3 Drought Identification through Standardized Drought Indices

A variety of drought indices has been proposed over the past decades to reduce the complex problem of drought identification
and quantification to single numbers (WMO and GWP, 2016). To identify spatiotemporal anomalies (i.e., drought periods) across the different hydrological components within the water cycle (e.g., meteorological, soil moisture, or hydrological drought in Figure 1), we selected the following widely-used standardized drought indices that quantify droughts as dimensionless deviations from normal conditions of the available hydroclimatic variables:

- The **standardized precipitation index (SPI)**, introduced by McKee et al. (1993), relies solely on *precipitation* and
has been a popular index for drought intercomparison projects due to its wide applicability for different spatiotemporal scales. It has been endorsed by the World Meteorological Organization (WMO) and others as the standard way of quantifying meteorological drought (WMO and GWP, 2016).





- The **standardized soil moisture index (SSMI)**, as described by Sheffield and Wood (2007), is derived from volumetric *soil moisture* values. These values echo the combined effects of a range of hydrological processes, such as plant transpiration, soil evaporation, infiltration, runoff, and the accumulation and melting of snow.

- The **standardized streamflow index (SSFI)** operates on a similar concept as the previous indices, but utilizes *streamflow* data instead (Vicente-Serrano et al., 2011). It is commonly used to quantify hydrological droughts (Vicente-Serrano et al., 2012).

- The **standardized groundwater index (SGI)**, developed by Bloomfield and Marchant (2013), standardizes *groundwater level* time series to characterize groundwater droughts. Given the complexity of groundwater flow systems, the SGI is useful for understanding fluctuations in groundwater levels and storage in response to variations in water input.

All indices were computed by fitting a probability distribution function to the monthly series of the relevant variable to estimate cumulative probabilities, which were then transformed into so-called z-scores based on a normal distribution with zero mean and a standard deviation of one. The standardized index then equaled these z-scores, which implies that it represented the number of standard deviations away from the mean. Positive values indicate conditions above normal (no drought), while negative values represent below-normal conditions (extreme drought ≤ -2, severe ≤ -1.5, moderate ≤ -1.0 and mild drought ≤ 0). All indices were calculated for different aggregation periods (i.e., for 1, 3, 6, 12 and 24 months), following the procedures described in Teutschbein et al. (2022). Shorter aggregation periods (< 3 months) capture short-term fluctuations and help to understand immediate impacts, while longer aggregation periods (> 6 months) reflect longer-lasting anomalies and accumulating effects through the water cycle.

The computation of all standardized indices is influenced by both the data record length (Wu et al., 2005) and the selected probability distribution function (Stagge et al., 2015). We followed the methodology outlined by Teutschbein et al. (2022) and tested 16 one-, two and three-parameter distributions probability distributions (Figure 3) for each considered series (4 indices, obtained for 50 gauging stations and 12 months of the year) separately. For a detailed description and corresponding equations of each tested distribution, we refer the reader to Teutschbein et al. (2022). The Kuiper's goodness-of-fit test (Kuiper, 1960) was utilized to select the best-fitting distribution, which was then transformed into z-scores (i.e., a normal distribution with zero mean and a standard deviation equal to 1), which directly provided the standardized index values in each catchment, for each calendar month and each aggregation period.



**Figure 3: Summary of 16 candidate distributions and the percentage of times each was selected as the best fit for 3-month aggregated data across the four hydrologic variables. The final column shows the overall selection frequency across all fitting instances. Best-fitting distributions are highlighted in varying shades of purple, with darker shades indicating higher selection percentages.**

## 2.4 Detection of Spatiotemporal Patterns

We analyzed general spatiotemporal patterns in droughts by considering the standardized drought indices (obtained from the 3-month and 12-month aggregation periods) jointly over the entire record period (1961–2020), and across all catchments. As a result, we were able to identify drought periods, and to analyze their spatial prevalence. Special emphasis in our analyses was put on evaluating potential north-south patterns, which we expected to emerge due to variations in evaporation as well as snow accumulation and snow melt.

## 2.5 Statistical Analysis of Drought Propagation

For all statistical analyses of drought propagation described below, we adopted the 3-month aggregation period, which is regularly considered for assessment of drought impacts on freshwater ecosystems, water supply, hydropower production and industry (Bae et al., 2019; Stagge et al., 2015). The choice of a 3-month period strikes a balance between capturing the immediate effects of precipitation deficits and the broader, seasonal impacts.





### 2.5.1 Propagation Time

Given the inherent complexity of hydrological processes and their interactions, considerable variability in drought responses is expected across different hydrological components. To assess how various components react to precipitation deficits, we calculated the cross-correlation between the SPI and the three other indices (SSMI, SSFI, and SGI), focusing exclusively on periods of moderate, severe or extreme drought (i.e. indices < -1), thus excluding mild droughts and non-drought conditions from the analysis. Cross-correlation served as a metric to quantify the similarity between the SPI time series and the lagged time series of the other indices, with lags ranging from 1 to 12 months.

Following the procedure outlined by Bloomfield and Marchant (2013), which was initially applied to groundwater response times, we used the lag time that produced the highest Spearman's rank correlation coefficient $\theta_s$ (Spearman, 1904) as an indicator of response times. Lag time is a commonly used metric to describe drought propagation (Zhang et al., 2022), where shorter lag times indicate a faster response of a specific component to SPI changes.

Propagation times were determined separately for each catchment, and the results were grouped and analyzed according to catchment clusters (c.f. Figure 2c) to facilitate the identification of spatial patterns.

### 2.5.2 Propagation Probability

To investigate the role of precipitation deficits in triggering drought conditions across the hydrological system, we conducted a probabilistic analysis focused on the occurrence of different drought types following a precipitation drought event. After identifying precipitation droughts with SPI < -1 (i.e., excluding mild droughts) as separate events/runs, we evaluated whether corresponding deficits in soil moisture (SSMI), streamflow (SSFI) or groundwater (SGI) could be observed within different drought propagation time frames. This analysis is captured by the conditional probability expressed in Eq. (1).

$$P\left(index_{lag} < -1 \mid SPI < -1\right) = \frac{P\left(SPI < -1 \cap index_{lag} < -1\right)}{P(SPI < -1)} \tag{1}$$

Here, $index_{lag}$ represents any of the drought indices for soil moisture, streamflow, or groundwater (i.e., SSMI, SSFI or SGI) at a given temporal $lag$. Therefore, $P\left(index_{lag} < -1 \mid SPI < -1\right)$ is the conditional probability of any of the drought indices indicating drought conditions given that a precipitation drought (indicated by SPI < -1) has already occurred.

In this analysis, we tested all possible time lags ranging from 1 to 12 months. This involved starting with the probability that a precipitation drought would cause a soil moisture, streamflow, or groundwater drought within the same month, then extending the analysis to include the same and subsequent months, and so forth, up to 12 months.

Additionally, we conducted this analysis separately for precipitation deficits occurring in different seasons, where (1) spring correspondents to the months of March, April and May (MAM), (2) summer includes the months June, July and August (JJA), (3) autumn covers September, October and November (SON), and (4) winter includes the months of December, January and February (DJF). To assess the variation of clusters within each season, we utilized the coefficient of variation (CV).





### 2.5.3 Governing Factors of Drought Propagation

To explore the relationship between propagation time and probability with underlying catchment characteristics and hydroclimatic factors, we employed the non-parametric Spearman's rank correlation coefficient $\theta_s$ (Spearman, 1904). This statistical approach allowed us to assess correlations across all pairwise combinations of standardized drought indices and a comprehensive range of potential influencing factors. These factors included geographic and physical attributes such as latitude, catchment area, elevation, and slope, as well as hydrological and climatic variables like the degree of regulation (DOR), regulation volume, annual mean precipitation, annual mean temperature, and annual streamflow. Additionally, we considered the impact of soil types and land cover variations, such as the percentage of forest cover or agricultural land.

The choice of Spearman's rank correlation over the linear Pearson product-moment correlation (Pearson, 1920) was driven by its ability to capture monotonic relationship without assuming any specific form of the relationship, such as linearity or logarithmic behaviour. This flexibility is particularly advantageous for drought propagation through the hydrological system, because relationships between variables often do not follow a strict linear pattern (Mukherjee et al., 2018). By using Spearman's correlation, we ensured that our analysis could detect and quantify both linear and non-linear associations, providing a more robust understanding of how different catchment and climatic characteristics influence drought propagation dynamics.

### 2.6 Droughts and their Propagation in a Changing Climate

The extensive 60-year observational data record encompassed two 'climate normal periods' (CNPs), each representing a 30-year average of the Earth's climate, as defined by the World Meteorological Organization (WMO Climatological Normals | World Meteorological Organization, 2021). This provided a unique opportunity to conduct an assessment of potential long-term shifts in droughts and their propagation in high-latitude catchments. Specifically, we investigated

whether there were significant differences in both the drought indices, the propagation time and the propagation probability in each catchment cluster between the CNPs of 1961-1990 and 1991-2020, utilizing the non-parametric Wilcoxon rank sum test (Asadzadeh et al., 2014; Montgomery and Runger, 2010). This analysis focused primarily on precipitation drought propagation into soil moisture and streamflow, as the available groundwater observations were not of sufficient length for such an assessment.

To analyze variations in annual drought conditions, we utilized 12-month aggregated drought indices (SPI-12, SSMI-12, SSFI-12) for the month of September of each year. This selection covers the full water year, from October of the previous year to September of the present year. For seasonal drought conditions, we employed 3-month aggregated drought indices (SPI-3, SSMI-3, SSFI-3) corresponding to the final month of each season: May (spring, MAM), August (summer, JJA), November (autumn, SON), and February (DJF).





**Figure 4:** Computed standardized indices for (a-b) precipitation (SPI), (c-d) soil moisture (SSMI), (e-f) streamflow (SSFI) and (g-h) groundwater (SGI) during the period 1961–2020 for all Swedish study catchments sorted by latitude (y-axis) over time (x-axis) and over two different aggregation periods: left panels represent the 3-month indices, the right panels the less-noisy 12-month indices. White colors indicate no drought conditions, while gray highlights only mild drought conditions. Darker red and blue colors indicate a more severe drought than lighter yellow colors. For SGI (g-h), the black colors represent missing values.




## 3 Results

### 3.1 Spatiotemporal Drought Patterns across the Hydrological System

270 The computed standardized indices (SPI, SSMI, SSFI, and SGI) at 3-month aggregation period highlight multiple significant drought periods across the country, with both temporal and spatial variations evident across the different hydrological components (Figure 4, left panels). We also showcase the less noisy 12-month aggregation period to facilitate visual comparisons of different catchments and drought indices (Figure 4, right panels). Notably, two pronounced drought periods, occurring in the late 1960s and mid-1970s, affected the entire country and had major impacts on the hydrological system.

275 These events began with precipitation deficits, as indicated by the SPI (Figure 4a,b), and propagated through to soil moisture (Figure 4c,d), streamflow (Figure 4e,f), and groundwater systems (Figure 4g,h), with each component experiencing drought conditions for several months.

The droughts of 1996 and the 2003 European heatwave are also clearly visible across all hydrological components, demonstrating the widespread effects of these events. Similarly, more recent droughts, particularly those in 2016/17 and 2018, 280 are well represented, affecting soil moisture, streamflow, and groundwater, although the severity and extent vary by region.

Beyond these general patterns, the standardized indices also reveal notable spatial variations in how different hydrological components were affected by droughts. For example, the early 1990s drought primarily affected catchments in southern Sweden, where the impact on soil moisture (Figure 4c,d) was substantially stronger compared to other components. Similarly, the drought of 2016/17 and 2018 display distinct spatial patterns: While precipitation and streamflow deficits were observed 285 nationwide, groundwater levels were disproportionately affected in southern Sweden.

### 3.2 Propagation Time

The cross-correlation analysis uncovered differences in drought propagation times across the five identified clusters (Figure 5). Soil moisture exhibited the shortest propagation times across all clusters, with average response times ranging from 0.1 to 2.7 months, highlighting its sensitivity to short-term climatic fluctuations. Streamflow showed slightly longer propagation 290 times compared to soil moisture, with some regional differences observed. The western clusters 1, 3 and 4 (Figure 5a,c,e.) exhibited response times of 0.9 to 2.4 months, whereas the other two clusters - Cluster 2 (Figure 5b) and Cluster 5 (Figure 5d) - had somewhat longer average response times of 3 and 3.5 months, respectively. This suggests that while streamflow responds quickly to precipitation deficits, the speed of this response is influenced by regional hydrological and climatic conditions.

Groundwater displayed the longest and most variable propagation times among the hydrological components analyzed. The 295 response times varied considerably across clusters, with slightly longer response times in western clusters 2 and 5. Note that the northernmost Cluster 1 (Figure 5a) had only one groundwater observation in the region, for which no correlation could be found to precipitation deficits. In the other clusters, average groundwater response times ranged from 2.6 months to 6.7 months. Groundwater also exhibited the highest internal variability within each cluster, with differences of several months between the fastest and slowest responses. Clusters 2 and 5 were particularly notable, with differences of up to 10 months observed between





300  the shortest and longest groundwater response times. This high variability underscores the complexity of groundwater dynamics and its longer, more delayed response to precipitation deficits compared to other hydrological components.

Across all 50 catchments in Sweden (Figure 5f), soil moisture responds to precipitation deficits within an average in one month, streamflow in two months, and groundwater within 4 months.

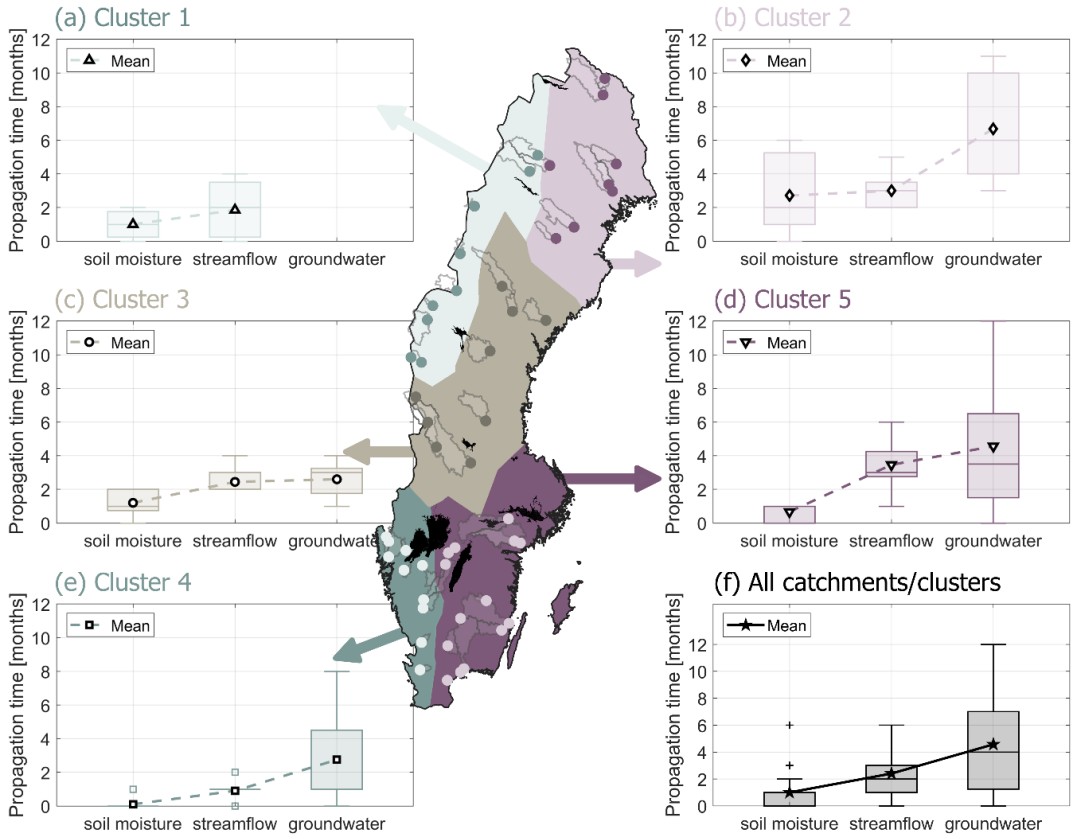

305

**Figure 5: Propagation times (in months) from precipitation deficits to soil moisture, streamflow and groundwater drought across the five different catchment clusters (a-e) and across all catchments (f) in Sweden.**

### 3.3 Propagation Probability

To quantify how precipitation deficits contribute to the development of more severe droughts in other hydrological
310  components, we analyzed conditional probabilities. The systematic assessment of step-wise time lags (Figure 6) reveals that - at the shortest lag of one month - a precipitation deficit results in an agricultural/ecological drought in 48% of cases, a streamflow drought in 41%, and a groundwater drought in 38% of cases (Figure 6a). As the lag times are progressively extended month by month, the likelihood of a precipitation drought triggering subsequent droughts in other components increases considerably. Over time, the frequencies of agricultural/ecological, streamflow, and groundwater droughts converge,
315  indicating a more uniform occurrence of drought across the hydrological system.





**Figure 6: Probability of a moderate, severe, or extreme precipitation drought (SPI < -1) triggering subsequent droughts in soil moisture, streamflow, or groundwater (SSMI, SSFI, or SGI < -1) across different time lags. Panel (a) shows the overall conditional probability for all catchments combined. The boxplots depict the distribution of probabilities across catchments, with the solid black line representing the median probability for a 1-month lag time. The thin grey lines correspond to the medians probabilities for stepwise increasing time lags, while the lines for 3 and 12 months are highlighted for clarity. The bottom panels provide a detailed breakdown by cluster for (b) soil moisture (SSMI), (c) streamflow (SSFI), and (d) groundwater (SGI), respectively.**

Across the clusters, the conditional probability of a soil moisture drought occurring within one month of a precipitation deficit ranges from 41 to 43% in the northern clusters 1-3, increasing to 51% and 57% in the southern clusters 4-5. As the time lag increases, the probabilities rise, and the differences between northern and southern catchments become less pronounced. This general pattern also applies to streamflow (Figure 6c) and groundwater droughts (Figure 6d). However, these components do not exhibit as strong a north-south gradient. Instead, the propagation probabilities across the different clusters are more balanced, with the exception of cluster 4, located in southwestern Sweden, where the probabilities of precipitation deficits propagating to streamflow or groundwater consistently remain 20-30% higher than in other Swedish regions.





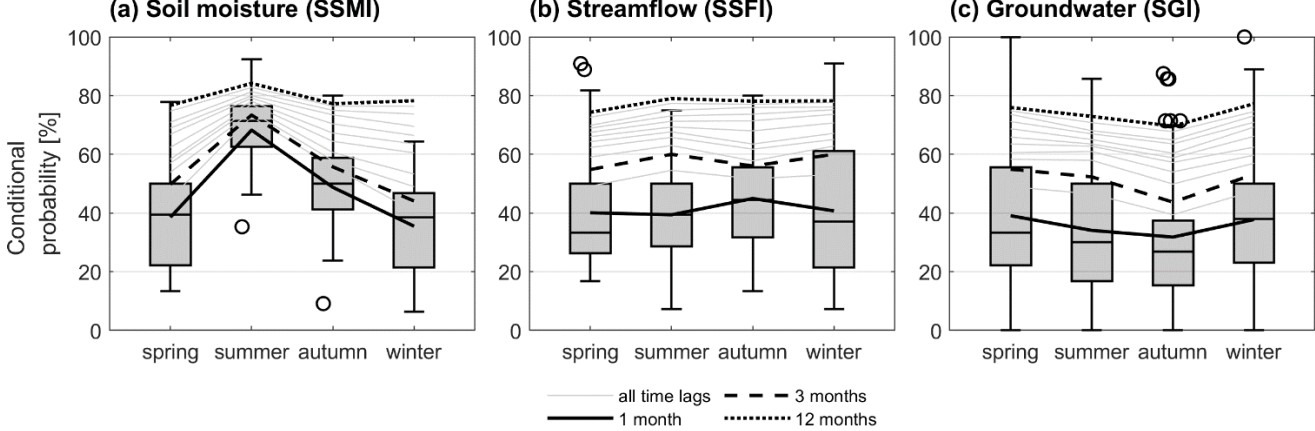

**Figure 7: Probability of a moderate, severe, or extreme precipitation drought (SPI < -1) in different seasons (spring, summer, autumn and winter) triggering subsequent droughts in (a) soil moisture, (b) streamflow, or (c) groundwater (SSMI, SSFI, or SGI < -1) across different time lags. The boxplots depict the distribution of probabilities across catchments, with the solid black line representing the median probability for a 1-month lag time. The thin grey lines correspond to the medians probabilities for step-wise increasing time lags, while the lines for 3 and 12 months are highlighted for clarity.**

The propagation probability did not only vary across clusters, but also across seasons (Figure 7). The biggest difference can be seen in the propagation probability to soil moisture (Figure 7a), where the highest propagation probabilities occur in summer. This means, a precipitation deficit of SPI < -1 in summer has on average a 68% probability to entail a soil moisture drought within one months, while the probability in spring and winter is less than 40%. This pattern is consistent also for longer time lags. For streamflow, the pattern is less pronounced. At a 1-month time lag, the propagation probability is slightly higher in the autumn (45%) than in the other seasons (~40%, Figure 7b). However, when considering longer time-lags, spring also consistently shows somewhat higher propagation probabilities (Figure 7b). For groundwater, propagation probabilities remain fairly consistent – ranging from 34% to 39% in spring, summer and winter (Figure 7c). Only autumn sticks out with somewhat lower probabilities (32% at 1-month lag), a pattern that persists across all time lags.





**Figure 8: Cluster-wise probability of a moderate, severe, or extreme precipitation drought (SPI < -1) triggering subsequent droughts by season (rows) and type of subsequent drought (columns). Each subplot also depicts the coefficient of variation (CV) across clusters for time-lag 1.**

350




A more detailed analysis of propagation probability by cluster reveals distinct north-south patterns during certain seasons and drought types (Figure 8). In spring (MAM), the propagation probability to soil moisture (Figure 8a) is considerably higher for the two southernmost clusters (4 and 5) at 50% and 53%, compared to the northern clusters, where it ranges from 23% to 27%). For streamflow (Figure 8b) and groundwater (Figure 8c), cluster 4 stands out with notably higher propagation probabilities exceeding 60%.

In summer (JJA), the propagation probability for soil moisture increases linearly from 60% to nearly 75% across all time lags as one moves further south (Figure 8d). Conversely, the propagation probability to streamflow (Figure 8e) shows an inverse trend at a 1-month time lag, i.e., decreasing from 47% in the north to 30% in the south, with the exception of cluster 4 (53%). This north-south gradient is less pronounced for longer time lags.

In autumn (SON), the spatial patterns resemble those of summer. However, the overall propagation probabilities to soil moisture are 10% to 20% lower than in summer (Figure 8g), while probabilities for streamflow (Figure 8h) and groundwater (Figure 8i) are slightly higher compared to summer levels.

Winter (DJF) exhibits the lowest propagation probabilities to soil moisture (Figure 8j) across all clusters, with the exception of cluster 3, which had the lowest probability in spring. Both winter and spring display the highest variations in soil moisture (Figure 8a,j) and streamflow (Figure 8b,k) across clusters, with coefficients of variation (CV) ranging from 0.34 to 0.44. Groundwater showed generally higher variations than both soil moisture and streamflow, with CVs ranging from 0.42 to 0.59, peaking in summer (Figure 8f) and winter (Figure 8l).

## 4 Governing Factors of Drought Propagation

Among the three hydrological components, the propagation time to soil moisture generally exhibits the most significant correlations with physical/topographic catchment features (Figure 9a, left). Five features - three classified as catchment characteristics and two as land-cover features - show significant positive correlations (p-value < 0.05): latitude ($\theta_s = 0.42$), elevation ($\theta_s = 0.31$), slope ($\theta_s = 0.33$), shrubs and grassland ($\theta_s = 0.41$), and wetlands ($\theta_s = 0.43$) are associated with longer propagation times. Significant negative correlations are found for five features, including annual mean precipitation ($\theta_s = -0.34$), annual mean temperature ($\theta_s = -0.37$), silt ($\theta_s = -0.44$), urban areas ($\theta_s = -0.42$) and agriculture ($\theta_s = -0.41$). For propagation time to streamflow (Figure 9a, center), significant positive correlations are observed with catchment area ($\theta_s = 0.31$), till soils ($\theta_s = 0.41$), and water features ($\theta_s = 0.29$). Conversely, annual precipitation and streamflow exhibit strong negative correlations with drought propagation time ($\theta_s = -0.59$ and $-0.49$, respectively), indicating that higher precipitation and streamflow generation are associated with shorter propagation times. Propagation times to groundwater tend towards negative correlations (Figure 9a, right), with significant correlations observed only for annual precipitation ($\theta_s = -0.75$) and areas annual streamflow ($\theta_s = -0.56$). Notably, only one single catchment characteristic exerts a strong influence on drought propagation across all three hydrological components simultaneously, namely annual precipitation, indicating shorter




propagation times for catchments with higher annual precipitation. This highlights the distinct factors driving drought dynamics and propagation times to soil moisture, streamflow, and groundwater.

In terms of propagation probabilities, numerous significant correlations with physical/topographic catchment features are seen for soil moisture (Figure 9b, left). Negative correlations are detected with all catchment features, particularly with latitude ($\theta_s$ = - 0.67), elevation ($\theta_s$ = - 0.73) and slope ($\theta_s$ = - 0.57). Strong negative correlations are also found with till or weathered deposits ($\theta_s$ = - 0.61), open land ($\theta_s$ = - 0.53), shrubs and grassland ($\theta_s$ = - 0.60) as well as wetlands ($\theta_s$ = - 0.51). In contrast, strong positive correlations are observed with annual temperature ($\theta_s$ = 0.81) and agriculture ($\theta_s$ = 0.68). For propagation time to streamflow (Figure 9b, center), fewer significant correlations are evident. Negative correlations are found with catchment area ($\theta_s$ = - 0.39), till soils ($\theta_s$ = - 0.50) and water bodies ($\theta_s$ = - 0.33), whereas strong positive correlations are detected for annual precipitation ($\theta_s$ = 0.73) and streamflow ($\theta_s$ = 0.59). Propagation time to groundwater shows only three significant correlations, all of them positive: annual precipitation ($\theta_s$ = 0.66) and streamflow ($\theta_s$ = 0.59), as well as clayey and clay till soils ($\theta_s$ = 0.66).

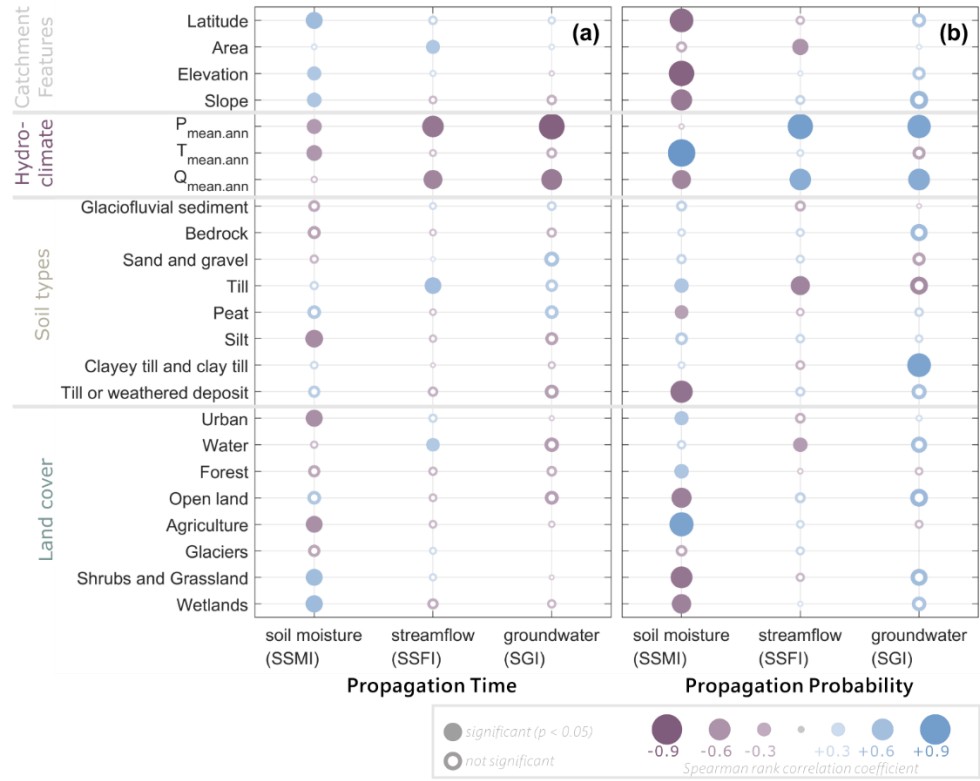

**Figure 9: Spearman's rank correlation coefficients between various catchment characteristics, ranging from physical/topographic catchment features, hydroclimatic properties, to soil types and land cover (y-axis), and (a) drought propagation times, as well as (b) propagation probabilities of precipitation deficits into soil moisture, streamflow and groundwater (x-axis). Larger/darker circles represent a stronger correlation coefficient, with blue indicating positive correlations and purple negative correlations. Filled circles represent significant correlations.**





**Figure 10: Shifts in (a-e) annual and (f-j) seasonal drought indices (SPI, SSMI and SSFI) between two climate normal periods, 1961–1990 (CNP1) and 1991–2020 (CNP2), for five hydrological clusters. Boxplots in panels (a)-(e) represent the range of index values (SPI, SSMI and SSFI) across catchments within each cluster, with significant shifts between CNP1 and CNP2 indicated by 'sign.' based on the Wilcoxon rank-sum test (p < 0.05). Panels (f)-(j) display the absolute change in indices for different seasons (spring, summer, autumn, and winter), with darker blue shading representing more pronounced increases in index values (wetter conditions), and darker purple shading representing stronger decreases (drier conditions). Significant seasonal changes are highlighted with black text.**





### 4.1 Droughts and their Propagation in a Changing Climate

**4.1.1 Changes in Spatiotemporal Drought Patterns**

The analysis of 12-month aggregated drought indices reveals a general upward trend from CNP1 to CNP2 (i.e., indices become more positive), indicating a reduction in the severity of droughts overall (Figure 10a-e). In all five clusters, the Wilcoxon rank-sum test consistently rejected the null hypothesis that the SPI and SSMI values for CNP1 and CNP2 were drawn from distributions with equal medians. This indicates a statistically significant shift ($p < 0.05$) in precipitation and soil moisture

drought conditions across Sweden between these two periods. For the SSFI, significant differences between CNP1 and CNP2 were only observed in the northernmost clusters 1-3 (Figure 10a-c), implying a notable shift in streamflow drought severity in these areas.

Seasonal analysis of 3-month aggregated drought indices shows a similar pattern, with predominantly positive shifts across most clusters and seasons (Figure 10f-j), suggesting a reduction in drought severity. However, some exceptions are observed.

In the two northernmost clusters 1 and 2, autumn indices (particularly soil moisture and streamflow) exhibit slight negative shifts, indicating slightly drier conditions during this season (Figure 10f, g). Similarly, in the southernmost clusters 4 and 5, substantial negative changes are observed in spring soil moisture and streamflow indices (Figure 10i, j), suggesting increased drought severity during this season in southern Sweden.

### 4.1.2 Changes in Propagation Time

For propagation time to soil moisture drought (SSMI) (Table 1a), most clusters exhibit no change between CNP1 and CNP2 (i.e., values of 0), except for cluster 2, which shows a modest increase of +0.5 months. For propagation time to streamflow drought (SSFI), cluster 2 also experiences a small increase (+0.5 months), while cluster 3 shows a more noticeable reduction of -1 month. However, in all cases, the Wilcoxon rank-sum test failed to reject the null hypothesis that the propagation times to soil moisture and to streamflow during CNP1 and CNP2 were drawn from continuous distributions with equal medians (Table

1a). This implies that there has been no statistically significant change (all p-values > 0.05) in the drought propagation times between these two periods.



Table 1: Drought Propagation Characteristics from precipitation deficits to soil moisture (SSMI) and streamflow droughts (SSFI), including the (a) observed median propagation times and (b) propagation probabilities for drought transitions. The changes in these characteristics are compared between two climate periods, CNP1 (1961-1990) and CNP2 (1991-2020). The p-values reflect the statistical significance of these changes, tested using the Wilcoxon rank-sum test. Significant changes (p-values <0.5) are highlighted in bold.

| Propagation Characteristics | SSMI | | | | SSFI | | | |
|---|---|---|---|---|---|---|---|---|
| | CNP1 | CNP2 | Change | p-value | CNP1 | CNP2 | Change | p-value |
| **(a)** **Propagation Time** | | | | | | | | |
| All catchments | 0 | 0 | 0.0 | 0,299 | 2 | 1 | - 1.0 | 0,197 |
| Cluster 1 | 0 | 0 | 0.0 | 1,000 | 1 | 1 | 0.0 | 0,734 |
| Cluster 2 | 0 | 0.5 | + 0.5 | 0,657 | 2 | 2.5 | + 0.5 | 0,835 |
| Cluster 3 | 0 | 0 | 0.0 | 0,056 | 2 | 1 | - 1.0 | 0,120 |
| Cluster 4 | 0 | 0 | 0.0 | NaN | 1 | 1 | 0.0 | 1,000 |
| Cluster 5 | 0 | 0 | 0.0 | 0,653 | 2 | 1.5 | - 0.5 | 0,135 |
| **(b)** **Propagation Probability** | | | | | | | | |
| within 1 month | 47% | 48% | + 1% | 0,197 | 42% | 41% | - 1% | 0,796 |
| within 2 months | 52% | 54% | + 2% | 0,081 | 53% | 52% | - 1% | 0,416 |
| within 3 months | 55% | 59% | **+ 5%** | **0,043** | 61% | 57% | - 4% | 0,591 |
| within 4 months | 58% | 64% | + 7% | 0,070 | 62% | 60% | - 2% | 0,828 |
| within 5 months | 61% | 66% | **+ 5%** | **0,048** | 64% | 63% | - 1% | 0,809 |
| within 6 months | 64% | 71% | **+ 7%** | **0,038** | 69% | 66% | - 3% | 0,836 |
| within 7 months | 68% | 72% | + 4% | 0,091 | 70% | 69% | - 1% | 0,728 |
| within 8 months | 71% | 75% | + 4% | 0,076 | 74% | 70% | - 4% | 0,661 |
| within 9 months | 73% | 75% | + 2% | 0,101 | 75% | 72% | - 4% | 0,735 |
| within 10 months | 75% | 77% | + 2% | 0,236 | 76% | 74% | - 3% | 0,715 |
| within 11 months | 79% | 80% | 1% | 0,291 | 78% | 74% | - 4% | 0,365 |
| within 12 months | 81% | 82% | 1% | 0,274 | 80% | 78% | - 2% | 0,336 |

### 4.1.3 Changes in Propagation Probability

Similarly, we investigated as to whether the conditional probability of a soil moisture respective streamflow drought following a precipitation drought has changed over time (Table 1b). The probability of propagation to soil moisture (SSMI) generally increased by 1% to 7% in CNP2 compared to CNP1, while the probabilities of propagation to streamflow (SSFI) slightly decrease by -1% to -4%. Despite these observed changes, the Wilcoxon rank sum test again did not reject the hypothesis of equal medians, indicating that no statistically significant shift occurred in drought propogation probabilities over the 60-year observational record (p-values > 0.05, Table 1b).





## 5 Discussion

### 5.1 Droughts across hydrological components

The computation of standardized drought indices enabled a visual exploration of the complex interactions between precipitation deficits and drought propagation across different hydrological components - soil moisture, streamflow, and groundwater - in high-latitude catchments. A consistent temporal alignment is observed between the standardized indices (SPI, SSMI, SSFI, and SGI), which is visually particularly clear for the longer 12-month aggregation period. This alignment suggests that prolonged precipitation deficits typically result in more severe and widespread drought conditions across all components,

affecting soil moisture, surface water, and groundwater.

Spatially, the analysis highlights notable variability, especially across latitudes. Southern regions, particularly during drought events like those in 2016 and 2018, experienced more pronounced impacts on groundwater systems (Bakke et al., 2020). This pattern may be attributed to unique regional hydrological processes, such as the role of snowmelt in the north versus higher evaporative demand, water extraction rates and slower recharge processes in the south. These findings emphasize the

importance of considering both spatial and temporal dimensions in drought analysis. They also suggest that groundwater systems in southern areas may be more vulnerable to prolonged dry periods, raising concerns about sustainable water management in these regions (Barthel et al., 2021).

While short-term indices capture the immediate impacts of precipitation deficits - critical for understanding the onset of drought conditions - the long-term indices reveal the persistence and severity of droughts that can strain water resources, ecosystems,

and agricultural productivity over extended periods (Stagge et al., 2015; Teutschbein et al., 2023b). Groundwater droughts, in particular, are less frequent but tend to be more severe and prolonged compared to those affecting soil moisture and streamflow (Bloomfield and Marchant, 2013). This highlights the need for sustained and proactive management strategies, especially in areas where groundwater serves as a critical resource.

### 5.2 Propagation Time

The analysis of drought propagation times across the five identified clusters revealed notable variations in response times among different hydrological components, specifically soil moisture, streamflow, and groundwater. These variations highlight the complex and region-specific nature of drought dynamics in the study area. Some general observations were that soil moisture consistently had the shortest propagation times, often responding almost immediately (within one month) to precipitation deficits. Therefore, soil moisture in the uppermost soil layer is highly sensitive to changes in precipitation, likely

due to its direct exposure to surface conditions (Singh et al., 2021). Streamflow generally exhibits slightly longer propagation times than soil moisture but remains within a relatively short timeframe, averaging about two months across all clusters. This indicates that while streamflow is also responsive to precipitation changes, the routing process through catchments introduces a slight delay compared to soil moisture (Robinson et al., 1995; Singh et al., 2021). In contrast, groundwater consistently displays the longest propagation times, averaging around four months, with significant variation observed across different



clusters. This delay in groundwater response is expected, as groundwater systems typically take longer to react to precipitation deficits due to the slower recharge of aquifers and the movement of water through subsurface layers. Notably, catchments with longer groundwater propagation times may have deep aquifers or slower recharge rates, leading to a delayed response (Gong et al., 2023). Our findings align with previous research indicating that groundwater responses to precipitation anomalies are more heterogeneous than those of streamflow (Weider and Boutt, 2010).

Clusters 3, 4, and 5 demonstrate shorter groundwater propagation times compared to clusters 1 and 2, indicating more responsive groundwater systems. This could be attributed to shallower aquifers or regions where groundwater recharge processes occur more rapidly (Cochand et al., 2020; Gong et al., 2023).

The variability in groundwater response across these clusters aligns with the findings by Bloomfield et al. (2013), Kumar et al. (2016), and Stoelzle et al.(2014), who found that the timescales of drought propagation into groundwater are highly site-

specific. This underscores the critical role of hydrogeological characteristics and subsurface storage processes in shaping groundwater responses, highlighting the necessity for region-specific drought management strategies. For instance, the consistent short propagation times for soil moisture and streamflow across all clusters emphasize the importance of rapid response measures for these components during drought events, as they quickly reflect precipitation deficits. In contrast, the longer and more variable propagation times for groundwater suggest that regions with delayed groundwater responses, such

as clusters 1 and 2, may require more proactive groundwater management. This could involve targeted monitoring and conservation measures to mitigate the impact of prolonged droughts on groundwater resources (Moore and Schindler, 2022; Thomann et al., 2022).

### 5.3  Propagation Probability

The analysis of conditional probability serves as a valuable tool for understanding how precipitation deficits influence the

broader hydrological system, acting as an indicator of the strength and transmission of drought signals into various regional hydrological components (Zhu et al., 2021). Specifically, it highlights how precipitation shortages can trigger subsequent droughts in other hydrological systems over different time lags. At the shortest lag (one month), the probability of a precipitation deficit leading to a soil moisture drought is the highest, emphasizing that agricultural and ecological systems are often the first to be impacted by water shortages. In contrast, the probabilities of precipitation deficits propagating to

streamflow and then to groundwater are lower (Geng et al., 2024b; Meresa et al., 2023; Wang et al., 2022).

As the lag time systematically extends, the conditional probabilities for all three components (soil moisture, streamflow, groundwater) begin to converge, indicating that prolonged precipitation deficits increase the likelihood of all components experiencing drought. This suggests that while some systems (like soil moisture) respond quickly, over time, the effects of prolonged precipitation deficits will eventually impact all components, necessitating a comprehensive approach to drought

risk management, which considers the entire cycle of disaster management from prediction and prevention to practical measures reducing impacts of droughts and supporting recovery in a sustainability context (AghaKouchak et al., 2015; Grobicki et al., 2015).



There is clear spatial variability in how droughts propagate across different clusters, with distinct north-south gradients observed in the propagation probabilities of soil moisture droughts. The fact that only 41%-43% of all precipitation deficits (SPI < -1) in the northern clusters 1-3 lead to soil moisture deficits within one month suggests that also other factors play a crucial role for the development of agricultural/ecological drought. Snow might be a key player, as is indicated by the probability getting higher when moving towards southern catchments (less snow, more rainfall). We can also speculate that this regional pattern is formed by other factors such as high soil water storage capacities (Geris et al., 2015) or different drought-generating mechanisms in these snow-dominated regions (Van Loon and Van Lanen, 2012). Conversely, clusters 4 and 5 exhibit higher probabilities (up to 57% within one month), indicating a stronger link between precipitation deficits and soil moisture droughts. These regions may have soil types or land cover that are more susceptible to rapid drying during periods of low precipitation, or may be influenced by regional climatic such as higher evaporative demand (Teutschbein, 2024a). However, this gradient is less pronounced for streamflow and groundwater, where probabilities are more balanced across clusters, except for cluster 4 in southwestern Sweden, which consistently shows higher conditional drought probabilities across all components. In fact, catchments in cluster 4 receive considerably higher amounts of annual precipitation (944 mm compared to the Swedish average of 784 mm), but also experience greater evaporative demand (particularly during summer) due to their southern location, while also receiving the least amount of snow (10-20% of annual precipitation compared to 30% in central and 40% in northern Sweden).

The delayed but persistent response of groundwater (propagation probabilities of 52-70% within 3 months in clusters 2-5) underscores the need for long-term monitoring and proactive management strategies, especially in regions with high groundwater dependence (Saito et al., 2021; Thomann et al., 2022). The variability in drought propagation probabilities and response times across different clusters and hydrological components suggests that a one-size-fits-all approach to drought management is unlikely to be effective across diverse regions and sectors (Stenfors et al., 2024c). Instead, a more tailored approach is required, involving context-specific analysis that considers local climatic, topographic, and hydrological conditions (Kchouk et al., 2022), while also recognizing the dependency of water supply on different resources, such as surface or groundwater (Stenfors et al., 2024b).

It is noteworthy that groundwater droughts featured the highest variability within clusters, which highlights the complexity of subsurface hydrological processes and may reflect differences in regional hydrological conditions, groundwater storage capacity, or data quality (Kumar et al., 2016; Stoelzle et al., 2014). Thus, more research efforts are needed to study the propagation into groundwater levels (Barthel et al., 2021) and link it to aquifer properties and recharge rates.

Our analysis also revealed significant seasonal variations in the probability of precipitation deficits leading to droughts in different hydrological components. Generally, summer stands out as the season with the highest propagation probabilities across all time lags, particularly for soil moisture droughts. This can be attributed to several factors: high evapotranspiration rates, reduced soil moisture recharge, and greater demand for water by vegetation during this period (Andersson, 1989; Cienciala et al., 1999), which make the soil system even more sensitive to precipitation deficits. In contrast, spring and winter show lower propagation probabilities to soil moisture droughts, likely due to lower evapotranspiration demands and a greater



influence of snow and snowmelt, which seems to buffer the effects of precipitation deficits and dampen soil moisture droughts (Potopová et al., 2016). These patterns are less pronounced for streamflow and groundwater droughts, which show relatively stable propagation probabilities across all seasons.

North-south differences emerged also in seasonal propagation probabilities, particularly for soil moisture and streamflow. While the overall inter-cluster patterns remain consistent across all four seasons, the absolute probabilities vary. For instance, the conditional probability of soil moisture droughts is generally higher in southern catchments and lower in northern catchments throughout the year. In spring, snowmelt plays a key role in mitigating precipitation deficits, resulting in significantly lower propagation probabilities in the snow-dominated northern clusters (1-3) of around 20-30% - compared to

50% in the southern clusters. During summer and autumn, total precipitation and evaporative demand become the dominant factors (Koster et al., 2019; Wang et al., 2022),, driving up the probability of soil moisture droughts, especially in the south. In winter, snow accumulation acts as a buffer, further reducing the propagation probabilities for soil moisture droughts, particularly in regions with substantial snowfall (Potopová et al., 2016). Conversely, streamflow propagation in summer decreases as one moves south, in line with the southwards-decreasing runoff coefficients (i.e., proportionally less precipitation

turns into runoff).

It is noteworthy that inter-cluster variation is largest during spring and winter for both soil moisture and streamflow propagation probabilities. This indicates that northern and southern catchments behave more distinctly in these seasons, with northern clusters showing stronger buffering effects due to snow accumulation and snowmelt. In contrast, summer and autumn see a more uniform response across regions, which emphasizes the proportionally larger influence of snow processes in the north

during spring and winter, compared to evaporative demand in the summer and autumn. However, groundwater propagation exhibits consistently high inter-cluster variability across all seasons, suggesting that additional regional factors, particularly in cluster 4, may outweigh the effects of snow and evaporative demand. This could include regional geological differences, such as aquifer characteristics and groundwater storage capacities, which play a critical role in determining the speed and extent of drought propagation in groundwater systems.

These results suggest that drought propagation and resulting impacts are highly dependent on both seasonal timing and geographical location, reflecting the complex interactions between climate conditions, hydrological processes, and land characteristics, which can either accelerate (e.g., high evaporative demand) or delay drought propagation (e.g., snow-melt in snow-dominated catchments) (Koster et al., 2019; Potopová et al., 2016). In practical terms, this analysis highlights the need for seasonally adaptive water management strategies, especially in the southern clusters where the propagation probability to

soil moisture is elevated. It also points to the importance of long-term monitoring, particularly for groundwater, where slow but persistent drought impacts may be underestimated in the short term (Barthel et al., 2021). In restoration efforts or pilot monitoring programs, the regional variability revealed in these findings could guide targeted interventions, such as enhancing water storage in high-risk areas or focusing conservation efforts during high-risk seasons (Srivastav et al., 2021).



### 5.4 Governing Factors of Drought Propagation

Our analysis revealed that the processes driving drought dynamics vary considerably between soil moisture, streamflow, and groundwater. Propagation time of precipitation deficits to soil moisture in the studied catchments is influenced by a range of governing factors, while for streamflow and groundwater, only four and two significant factors, respectively, were identified. Generally, faster propagation times across all three hydrological components occur in catchments with high annual precipitation, warmer temperatures, and higher annual streamflow levels. This suggests that in regions with greater water input

and output, the hydrological system responds more quickly to changes in precipitation, resulting in faster drought onset but potentially quicker recovery from drought conditions as well. Additionally, soil moisture and streamflow propagation times are shorter in catchments with less till, more silt, and fewer water features, suggesting that water bodies can exert a buffering and delay function. For soil moisture, propagation time is further controlled by urban areas and agriculture, where surface runoff dominates, and is more rapid in warmer catchments at lower latitudes and elevations. Conversely, open land types (e.g.,

shrubs and grasslands, open land, and wetlands) tend to slow water propagation and promote water retention.

The propagation probability generally exhibits stronger correlations than propagation time. Our analysis showed that propagation probability to groundwater was significantly correlated with just three factors: higher annual precipitation, greater streamflow, and the presence of clayey till, which enhanced the likelihood of groundwater recharge. For streamflow, propagation probability is influenced by a few more factors. Catchments with higher precipitation and streamflow also tend to

exhibit greater propagation probabilities. In addition, smaller catchments, less till, and fewer water features are associated with higher streamflow propagation probability. The propagation to soil moisture was associated with the greatest number of governing factors. Specifically, catchments with higher mean temperatures, more agriculture, and lower latitude, elevation, and slope had the highest soil moisture propagation probabilities. Lower streamflow in these areas further amplified the likelihood of soil moisture response.

Across both drought propagation characteristics - time and probability - annual precipitation and annual streamflow emerged as the most influential drivers across all three hydrological components, making them useful proxies for drought forecasting. However, the remaining factors shaping drought propagation dynamics vary by component, reflecting the distinct physical and hydrological processes at play. Soil moisture is generally influenced by a broader array of factors, whereas streamflow is more directly linked to catchment size, the presence of till soils, and overall water availability. Groundwater is most strongly affected

by hydro-climatic variables and the hydrological regime.

This analysis underscores that drought propagation is not solely determined by catchment properties; rather, it is shaped by complex interactions between seasonal precipitation, evaporative demand, and snow dynamics (Van Loon et al., 2015). The timing and severity of the drought (Bae et al., 2019), the season (Meresa et al., 2023), location (Bevacqua et al., 2021), precipitation patterns (or lack thereof), as well as antecedent conditions/'memory effects' (Bales et al., 2018; Soulsby et al.,

2021) all can potentially important roles in determining whether a meteorological drought will evolve into more severe forms, such as agricultural/ecological or hydrological droughts. In high latitudes, snow becomes a critical factor in drought



propagation, particularly for longer aggregation periods. This is evident in northern regions, where reduced precipitation does not always lead to other types of droughts, likely due to the buffering effect of snowpack. Moreover, not all precipitation deficits necessarily progress into agricultural/ecological or hydrological droughts, as these deficits can be mitigated by the

landscape's buffering capacity (Maxwell et al., 2021). Conversely, not all agricultural/ecological or hydrological droughts are (solely) triggered by precipitation deficits; some result from extraordinarily high evaporative demands, lack of snowmelt, or a combination of several factors. This complexity highlights the need for tailored drought management strategies that consider the specific characteristics and vulnerabilities of each hydrological component (Stenfors et al., 2024a). For instance, areas prone to prolonged groundwater droughts due to specific soils may require focused groundwater management efforts, while

regions with large catchments or significant till soil coverage may need to prioritize streamflow management to mitigate drought impacts.

## 5.5 Drought Propagation in a Changing Climate

The analysis of changes in standardized drought indices (SPI, SSMI, and SSFI) across two climate normal periods (CNP1 and CNP2) revealed distinct regional and seasonal patterns in drought dynamics, which are crucial for understanding the evolving

drought risk in various clusters across the studied region. All clusters consistently indicate a wetting trend, i.e. drought conditions across all hydrological components have consistently become less severe. This trend is stronger in northern catchments, with clusters 1 and 2 featuring especially strong increases during the cold season when snowmelt and seasonal precipitation are likely contributing to the water surplus. This pronounced seasonal signal highlights the importance of snowpack and spring melt in driving the hydrological response in these northern clusters, and suggests that these regions may

be less prone to drought in the near term. The southern clusters 4 and 5 show significant decreases in both soil moisture (SSMI) and streamflow (SSFI) indices, despite relatively stable or even slightly increasing precipitation (SPI) during that season. This implies that the available water is not being retained in the system, potentially due to increased evaporation, reduced infiltration, or changes in land use such as urbanization and agriculture. These regions may be increasingly prone to hydrological drought, where water deficits in the soil and streams become more frequent and severe, affecting ecosystems,

agriculture, and water supplies. This trend is particularly concerning as it indicates that even if precipitation remains stable, water resources in these southern clusters are becoming less available for both natural ecosystems and human use.

The evaluation of the differences in drought propagation times and conditional probabilities of drought occurrence between CNP1 and CNP2 demonstrated that there has been no statistically significant change in drought propagation times to soil moisture and streamflow across all clusters. This implies that, despite global climatic changes and the associated shifts in

precipitation patterns, the response times of these hydrological components to droughts have remained consistent over the past 60 years. The failure to reject the null hypothesis (p-values > 0.05) implies that any observed differences in propagation times are not large enough to be statistically significant, and thus, the underlying processes governing drought propagation have likely remained stable across these two periods.





Similarly, the investigation into the conditional probabilities of soil moisture and streamflow droughts following a precipitation
drought reveals predominantly no statistically significant change between the two CNPs for the vast majority of the evaluated
time lags. This broad failure of the Wilcoxon rank sum test to reject the hypothesis of equal medians (p-values > 0.05) suggests
that the likelihood of a soil moisture or streamflow drought occurring after a precipitation deficit has not changed significantly
over the 60-year observational record (on an annual basis). This may be different for seasonal drought events, but is – due to
the limited number of drought events, owing to their rare nature – more difficult to test given the existing data set. The observed
stability in drought propagation times and probabilities suggests that the physical properties of the landscape, such as soil type,
vegetation cover, and catchment characteristics, have a more dominant role in controlling drought propagation than any
potential changes in climatic conditions between these periods. However, further research – particularly into seasonal dynamics
– is needed to unravel these links.

The absence of significant changes in both drought propagation times and the conditional probabilities of drought occurrence
over the two climate normal periods has important implications for drought management. Despite an observed increase in
average temperature by 2.2 degrees and a 20% increase in precipitation over the past 60 years (Teutschbein et al., 2022), these
climatic shifts have not resulted in significant alterations in drought propagation within the study period. Thus, historical data
and established models for predicting drought behavior based on past observations may still be relevant and reliable under
current climate conditions. This stability allows for a certain level of confidence in using past trends to inform future drought
preparedness and mitigation strategies. However, it is also important to recognize that while no significant changes have been
detected over the studied periods, ongoing climate change could still pose future challenges and potential alterations in drought
propagation in these high-latitude catchments (Teutschbein et al., 2023b). Continuous monitoring and periodic reassessment
of these trends will be necessary to detect any emerging shifts in drought dynamics that could impact water resource
management and planning.

**6 Conclusions**

This study highlights key patterns in drought propagation across high-latitude regions by analyzing both *propagation time*,
(how quickly precipitation deficits impact soil moisture, streamflow and groundwater) and *propagation probability* (the
likelihood that a precipitation deficit leads to drought in these components). Soil moisture responds fastest, with a typical lag
of one month, followed by streamflow at two months. Groundwater exhibits the longest and most variable response times,
averaging four months.

Precipitation deficits are a major driver of droughts in high latitudes, with probabilities highest for soil moisture (48%),
followed by streamflow (41%) and groundwater (38%) within the first month. These probabilities increase over time, showing
that prolonged deficits can lead to more widespread and severe droughts. However, not all droughts are driven solely by
precipitation deficits; factors such as high evaporation or lack of snowmelt also contribute, highlighting the complexity of
drought dynamics.



Although all catchments are experiencing a general trend of increased wetness throughout the year, driven by snowmelt and rising precipitation, southern catchments show heightened vulnerability to hydrological drought in the spring. This is largely due to increasing evaporative demand, which offsets stable or slightly rising precipitation levels during that season. Importantly, no significant changes in drought propagation times or probabilities were observed over the past 60 years, but

continuous monitoring and reassessment remain essential to detect any emerging shifts in drought dynamics that could impact water resource management in the future.

Drought dynamics in high-latitude regions are influenced by diverse factors, including local topography, hydroclimatic conditions, and soil properties. Soil moisture propagation is governed by catchment features, hydro-climate and land cover, while streamflow and groundwater are more closely linked to water availability.

By focusing on the unique conditions of high-latitude regions, this research has contributed to the broader scientific understanding of drought dynamics, offering new perspectives on the interactions between various hydrological systems during prolonged dry periods. Future studies should expand the geographical scope and explore seasonal and climate change impacts to further validate these findings and enhance their generalizability. As climate change intensified, proactive and region-specific strategies will be essential to safeguard water resources and the ecosystems that depend on them.

**7 Data availability**

The CAMELS-SE data that support the findings of this study are openly available at the Swedish National Data Service at https://doi.org/10.57804/t3rm-v029, SND-ID: 2023-173. Version: 1

**8 Author Contribution**

- Claudia Teutschbein: Conceptualization, Methodology, Formal Analysis, Investigation, Resources, Data Curation,
Writing – Original Draft, Writing – Review & Editing, Visualization, Supervision, Project administration, Funding acquisition.
- Markus Giese: Methodology, Writing – Review & Editing, Supervision
- Thomas Grabs: Conceptualization, Methodology, Writing – Review & Editing, Supervision
- Andrijana Todorović: Methodology, Data Curation, Writing – Review & Editing
- Roland Barthel: Conceptualization, Project administration, Funding acquisition.

**9 Competing Interests**

The authors declare that they have no conflict of interest.



## 10 Acknowledgements

We would like to express our sincere gratitude to Sabrina Atigui for her assistance with data preparation, initial work on
compiling relevant literature and helpful input during discussions. During the preparation of this work, the corresponding
author used ChatGPT to improve writing style and to check for grammar and spelling. The authors carefully reviewed, edited
and revised ChatGPT-generated content to their preferences, and take full responsibility for the content of the publication.

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
