# Peer review of "Drought propagation in high-latitude catchments: Insights from a 60-Year Analysis Using Standardized Indices"

_EGUsphere, 2024_

## Referee Comment (RC2)

The study analyses the propagation of meteorological to soil moisture, streamflow and groundwater droughts in 50 catchments selected from across Sweden. Drought propagation is analyzed by calculating lag times and propagation probabilities which are computed on standardized time-series of meteorological and hydrological variables. The study tries to address an important knowledge gap of understanding drought propagation in high latitude watersheds. The paper is very well written; the motivation and methodology are clearly stated, and results are presented in a logical sequence. The quality of figures in the paper is exceptionally good and there are several interesting findings from the study. However, I feel that there is a need to strengthen the discussion section of the paper. The authors have mostly discussed findings which are well-established in literature such as faster propagation of meteorological droughts to soil moisture and a more delayed response on streamflow and groundwater. The authors should try to highlight findings unique to the study region which have not yet been found in other regions. The analysis on the role of catchment properties also needs to be enhanced. The following are my detailed comments:

1. Section 2.3: The major novelty of the study, as highlighted by the authors, is to analyze drought propagation in high latitude catchments. But the authors have not highlighted the aspects of drought propagation which are unique to high latitude catchments. Despite the importance of snow-related processes, particularly in clusters 1 and 2, the study has been carried out using SPI - in a similar manner to drought propagation studies in lower-latitude regions. Snowmelt, which is an important source of streamflow and groundwater recharge in high latitude regions, is controlled by a complex interplay of temperature and precipitation. It would be great if the authors could include snow-related variables such as SWE into the analysis and bring out novel insights regarding its role in drought propagation, which remains an important knowledge gap.

2. Section 2.5.1: The authors have used lagged-cross correlations to analyze the lag times between meteorological droughts and other drought types. The authors mention that the correlations are being calculated only for drought periods. Does that mean correlations are being calculated only for periods with SPI<-1 or when SSMI/SSFI/SGI<-1 or both? Please clarify.

3. There are several interesting results in the study which have not been discussed in much detail.

   (1) The first of them being the high streamflow and groundwater drought propagation probabilities in cluster 4. While both cluster 4 and 5 are rainfall dominated, why are the probabilities so high for streamflow droughts in cluster 4 (Figure 6)? In L518-525, the authors have listed some factors which "could be" responsible for these differences such as soil water holding capacities but without much analysis. Soil type and land use land cover maps of the study region, if available, can be used to investigate the reasons for this observation. Also, why is there a significant difference between propagation probabilities of cluster 1 and 2 for streamflow and groundwater droughts in Figure 6?
   (2) Figure 7: While it is clear that the propagation probabilities for soil moisture droughts are highest in summer due to higher evaporative demand, the reasons for streamflow and groundwater drought propagation probabilities is not very clear. Why is propagation probability higher for streamflow and lower for groundwater in Autumn?

(3) Figure 8: While soil moisture propagation probabilities increase from north to south, the streamflow propagation probabilities decrease from north to south during summer season. Why?

The authors can explore the physical mechanisms underlying these interesting patterns observed in the statistical analysis. The authors may add maps of physical features such as soil type, vegetation, land use, runoff coefficients, baseflow index, snow fraction and geological features, if available, to better explain these patterns. Such analysis may reveal unique insights regarding drought propagation in high latitude watersheds.

4. In Figure 9, the authors use correlation analysis to understand the physical factors affecting propagation probabilities and lag times. Physical catchment features like soil type and land use are also considered. However, I wonder if this analysis makes sense for soil moisture droughts considering that soil moisture droughts have been analyzed using reanalysis dataset in this study. Reanalysis models do not have a very accurate representation of soil types and land use features. Thus, conclusions based on this analysis could be misleading.

Minor comments:

1. L494: I could not follow why places with delayed groundwater response would require more proactive management. Shouldn't it be reverse?
2. L515: Please change to "other factors also".
3. L556: Please delete the extra comma.
4. L600: How can annual rainfall and streamflow be used for drought forecasting? Please elaborate.

---

## Author Response (AR2)

UPPSALA
UNIVERSITET

Dear Giulia,

I sincerely apologize for overlooking your earlier recommendation to revise the methodology section to avoid self-plagiarism. This was an unintentional oversight on my part, and I truly appreciate your patience in giving us the opportunity to make the necessary changes.

We have now thoroughly revised the methodology section (particularly Sections 2.3 to 2.5) to address the issue. I'm honestly not sure how we missed it in the first place, but we have taken extra care this time to ensure it is appropriately rephrased.

I hope the revised manuscript now meets the required standards.

Kind regards,

Associate Professor

Claudia Teutschbein, on behalf of the (co-)author team

2025-04-22
* * *
Dear Giulia,

Thank you for handling our manuscript and for kindly granting the time extension, it was greatly appreciated.

We would also like to thank you and the two reviewers for the thoughtful and constructive feedback on our manuscript, titled "*Drought propagation in high-latitude catchments: Insights from a 60-Year Analysis Using Standardized Indices*" (ID: egusphere-2024-2742). We truly appreciate the time and effort you and the reviewers dedicated to assessing our work and providing valuable insights. Your feedback has been instrumental in enhancing the clarity and overall quality of the manuscript.

In the revised manuscript, we have done our best to address the reviewers' suggestions. In particular, we have clarified the novelty of the study in the introduction, improved the methodology section, condensed some of the overly detailed results, and streamlined the discussion to reduce repetition and remove unsupported statements.

All changes have been tracked using the Track Changes feature in Microsoft Word. Detailed, point-by-point responses to each reviewer comment are provided below (in blue), with references to line numbers in the revised version where all changes have been accepted.

We hope that you and the reviewers find the revisions satisfactory and that the manuscript is now suitable for publication in the Special Issue: Drought, society, and ecosystems (NHESS/BG/GC/HESS inter-journal SI).

Kind regards,

Associate Professor

Claudia Teutschbein, on behalf of the (co-)author team

2025-04-15

**Reviewer 1**

The manuscript is comprehensive, well written, and organized. The four research questions outlined in the introduction are properly addressed in the results and discussion sections, although the results and discussions can be more objective. The justification provided for novelty is the lack of knowledge of drought propagation processes in regions of high-latitude catchments such as Sweden. The figures are informative and well presented, with only minor comments for improvement. As a suggestion for future research, it would have been very interesting and novel to visualize the drought propagation results between drought types, drought events, and/or catchments or clusters with cascading Sankey plots or network graphs. The results section could be more objective by focusing on the most important findings, it is not necessary to report everything that can be read on the figures or tables. Therefore, I suggest focusing on the overall patterns and outliers for clarity. In the discussion section, I would suggest aiming mostly at answering the proposed research questions and describing how the main results are linked to targeted management practices in terms of drought propagation dynamics. Overly broad statements and generalizations in the discussion can be shortened or removed for conciseness. Apart from summarizing the most important results in Sweden and how they agree with previous literature, the discussion should also reflect on the research gap and highlight the differences against other regions in the world. How unique are drought propagation patterns in Sweden or in high latitudes in general? Please highlight and emphasize in the introduction and discussion sections what are the potential new insights and main findings from this paper regarding drought dynamics that have not yet been covered in the literature. Please also include some discussion on the limitations of the modelling approaches and assumptions. In summary, I consider that the scientific and presentation quality of this manuscript are excellent, and the scientific significance is good but could be improved by highlighting and distinguishing the knowledge contributions on drought dynamics from this application in Sweden. Therefore, I recommend accepting the manuscript with minor revisions.

**General Response:** We thank the reviewer for the thoughtful, constructive, and encouraging comments. We are pleased to hear that the manuscript is considered comprehensive, well-written, and of excellent scientific and presentation quality. We have revised the manuscript carefully to improve objectivity, clarity, and scientific significance, particularly regarding results and discussion - in line with the detailed comments provided by the reviewer. Below, we provide a detailed response to each point and describe the revisions made.

R1_1:    Line 154: Why was the SPI index chosen over the SPEI? The SPEI, which includes evapotranspiration, would have the advantage of considering temperature in its formulation. Increasing temperatures due to climate change tend to increase evapotranspiration, leading to raised drought effects that could counteract the wetting trend in Sweden. This process is not directly considered in the SPI or the other indices. As mentioned in Line 632, increased evaporation could potentially explain why available water is not being retained in some systems.

**Response:** The primary reason for selecting SPI over SPEI or other indices (e.g. SMRI -standardized melt and rain index - as suggested by reviewer 2) was our intention to base the analysis as much as possible on observational data, which we have now clarified in the introduction (lines 106-109). Calculating SPEI or SMRI requires incorporating pot. evapotranspiration or snowmelt, which introduces additional uncertainties (e.g., due to selection of a model, or due to parameter equifinality) and would also demand a more detailed seasonal analysis, because evapotranspiration is most relevant during warmer months, while snowmelt (as used in SMRI) plays a larger role in colder months. While we agree that such an approach could yield valuable insights, it would have significantly expanded the scope of the paper, which already presents a comprehensive set of analyses. We therefore chose to focus on tracing observed precipitation deficits through the hydrological system, without the added complexity. This decision is consistent with several previous large-scale drought propagation studies (Gong et al., 2023; Odongo et al., 2023; Sattar et al., 2019; Wang et al., 2022; Xu et al., 2023), and allows a clearer isolation of the precipitation signal as it propagates through soil moisture, streamflow, and groundwater. That said, we fully acknowledge the value of SPEI in capturing temperature-driven changes in evaporative

demand, especially in the context of climate change. We have now added a discussion of this limitation in the manuscript and recommend incorporating temperature-sensitive indices like SPEI or SMRI in future research, particularly for more targeted analyses of individual drought events (new section 4.6, lines 686-714).

R1_2: Line 202: Please justify why SPI is taken as the reference for propagation time and probability analysis instead of any of the three other indices.

**Response:** SPI was selected as the reference index for the propagation analysis because it directly represents the primary climatic driver (i.e., precipitation) and is widely used as a benchmark in drought propagation studies (Gong et al., 2023; Odongo et al., 2023; Sattar et al., 2019; Wang et al., 2022; Xu et al., 2023). Our specific aim was to trace how deficits in precipitation propagate through the hydrological system, affecting soil moisture, streamflow, and groundwater over time. This rationale has now been clearly stated in the introduction (lines 95-97).

R1_3: Line 205: Is this study the first expanded application of this approach for streamflow and soil moisture drought? If so, I suggest mentioning this and other possible methodological advancements in the introduction.

**Response:** Thank you for this suggestion. While several previous studies have applied similar correlation-based approaches, they typically focus on the propagation between only two drought types, for example from precipitation deficits to soil moisture (Geng et al., 2024), from precipitation to streamflow (Hellwig et al., 2022; Ma et al., 2021; Zhang et al., 2022) or from precipitation to groundwater anomalies (Bloomfield and Marchant, 2013). Only a few recent studies (e.g., Odongo et al., 2023) have adopted a more holistic perspective on drought propagation across multiple components of the hydrological cycle. Although the methodology itself is not entirely new, our study applies it in a particularly comprehensive and novel setting, simultaneously assessing drought propagation across four components: precipitation, soil moisture, streamflow, and groundwater. Importantly, this is done in a high-latitude context, which remains underrepresented in the literature. The novelty of our work lies primarily in the holistic application of this framework to northern catchments and the new insights it provides into drought dynamics under cold-climate conditions. We have clarified this point in the revised introduction, although we have chosen not to emphasize methodological innovation as the primary contribution of the study (see revised lines 85-90)

R1_4: Line 338: Why are probabilities for soil moisture drought propagation during summer so much more pronounced than for other seasons and other drought types?

**Response:** We assume that this is an effect of evapotranspiration, which is discussed in lines 563-568. Presumingly, higher evapotranspiration during summer increase the soil moisture sensitivity to precipitation deficits, which can explain the stronger propagation from meteorological to soil moisture droughts in this season.

R1_5: Line 467: Please elaborate and exemplify management strategies that are specific to long-term groundwater droughts as opposed to other drought types, highlighting the application of your findings into practice.

**Response:** Thanks for pointing out this chance to elaborate on management strategies. We added a few concrete examples, such as adjusting groundwater abstraction rates, implementing managed aquifer recharge, and measures for diversification and drought reserves (lines 510-513).

R1_6: Line 493: Please include examples from the literature of rapid response measures.

**Response:** We now cite literature examples including deficit irrigation, mulching, optimized seeding dates (in agriculture) and adaptive reservoir management or water-saving advisories (in water supply systems) as rapid response measures for short-term drought events (lines 514-518).

[Figure]

R1_7:     Lines 499-503: Those appear to be generic statements that would better fit the introduction section than discussion.

**Response:** These sentences have been moved to the methodology, where they better fit as part of the motivation for this approach (lines 247-250).

R1_8:     Figure 2: Please correct the following typo in the figure caption: 'tempereature'.

**Response:** Thank you for spotting this mistake! Typo corrected (Figure 2).

R1_9:     Figure 4: Why is the spacing of the latitudes (y-axis) inconsistent? Please readjust for equal spacing if this is unintentional. I suggest changing the colormap so that no data is shown as white instead of black for improved contrast and readability, as in Bloomfield and Marchant 2013 Fig. 11.

**Response:** The unequal spacing on the y-axis is intentional. The catchments are sorted by latitude, but they are not evenly distributed across Sweden. Some latitudinal bands contain a higher concentration of catchments, while others have only a few. As a result, the spacing reflects the actual distribution of catchments rather than uniform latitudinal intervals. We have clarified this in the figure caption to avoid confusion. We have also adjusted the colormap of figure 4.

R1_10:    Figure 6: For improved clarity, I suggest writing the names of the corresponding indices in front of the labels b), c) and d).

**Response:** To improve clarity and reinforce the connection between the upper and lower panels, we have adjusted the layout to better align the lower panels with the corresponding boxes in panel (a). Additionally, we added grey shaded background boxes to visually link them. Since the indices are already labelled in panel (a), we believe this updated layout makes the correspondence between panels clearer without needing to repeat the index names in the labels of (b–d).

R1_11:    Figure 10: Why is the presentation between annual and seasonal shifts different? It would have been more informative and consistent to show seasonal shifts also as boxplots.

**Response:** We agree that presenting the seasonal shifts as boxplots could have added consistency and potentially improved interpretability. However, we opted for the current visualization to maintain clarity and avoid overcrowding. Displaying seasonal shifts as boxplots for all clusters would have required 20 individual plots (5 clusters × 4 seasons), which we felt would reduce readability and make the figure overly complex. Therefore, we have chosen to retain the original figure layout without changes, in order to balance informativeness with visual simplicity.

**Citation**: https://doi.org/10.5194/egusphere-2024-2742-RC1

**Reviewer 2**

The study analyses the propagation of meteorological to soil moisture, streamflow and groundwater droughts in 50 catchments selected from across Sweden. Drought propagation is analyzed by calculating lag times and propagation probabilities which are computed on standardized time-series of meteorological and hydrological variables. The study tries to address an important knowledge gap of understanding drought propagation in high latitude watersheds.

The paper is very well written; the motivation and methodology are clearly stated, and results are presented in a logical sequence. The quality of figures in the paper is exceptionally good and there are several interesting findings from the study. However, I feel that there is a need to strengthen the discussion section of the paper. The authors have mostly discussed findings which are well-established in literature such as faster propagation of meteorological droughts to soil moisture and a more delayed response on streamflow and groundwater. The authors should try to highlight findings unique to the study region which have not yet been found in other regions. The analysis on the role of catchment properties also needs to be enhanced.

**General Response:** We sincerely thank the reviewer for the encouraging and constructive feedback on our manuscript. We are pleased to hear that the paper is considered well-written, methodologically sound, and that the figures and overall structure were appreciated. We are especially grateful for the thoughtful suggestions to strengthen the discussion and deepen the interpretation of our findings, which we have done in line with the detailed comments from the reviewer. Below we address each of the comments in detail and outline the revisions we made.

R2_1:   Section 2.3: The major novelty of the study, as highlighted by the authors, is to analyse drought propagation in high latitude catchments. But the authors have not highlighted the aspects of drought propagation which are unique to high latitude catchments. Despite the importance of snow-related processes, particularly in clusters 1 and 2, the study has been carried out using SPI - in a similar manner to drought propagation studies in lower-latitude regions. Snowmelt, which is an important source of streamflow and groundwater recharge in high latitude regions, is controlled by a complex interplay of temperature and precipitation. It would be great if the authors could include snow-related variables such as SWE into the analysis and bring out novel insights regarding its role in drought propagation, which remains an important knowledge gap.

**Response:** We appreciate the reviewer's observation regarding the need to highlight features unique to high-latitude catchments. We agree that snow processes play a significant role in drought dynamics in these regions. However, the primary reason for selecting SPI over SMRI (standardized melt and rain index) or other indices (e.g. SPEI as suggested by reviewer 1) was our intention to base the analysis as much as possible on observational data, which we have now clarified in the introduction (lines 106-109). Calculating SMRI or SPEI requires incorporating pot. evapotranspiration or snowmelt, which introduces additional uncertainties (e.g., due to selection of a model, or due to parameter equifinality), and would also demand a more detailed seasonal analysis, because snowmelt is most relevant during colder months, while evaporation (as used in SPEI) plays a larger role in warmer months. While we agree that such an approach could yield valuable insights, it would have significantly expanded the scope of the paper, which already presents a comprehensive set of analyses. We therefore chose to focus on tracing observed precipitation deficits through the hydrological system, without the

added complexity. This decision is consistent with several previous large-scale drought propagation studies (Gong et al., 2023; Odongo et al., 2023; Sattar et al., 2019; Wang et al., 2022; Xu et al., 2023), and allows a clearer isolation of the precipitation signal as it propagates through soil moisture, streamflow, and groundwater. That said, we fully acknowledge the value of SSMI in capturing temperature-driven changes in snowmelt and water input, especially in the context of climate change. We have now added a discussion of this limitation in the manuscript and recommend incorporating temperature-sensitive indices like SPEI or SSMI in future research, particularly for more targeted analyses of individual drought events (new section 4.6, lines 686-714).

R2_2: Section 2.5.1: The authors have used lagged-cross correlations to analyze the lag times between meteorological droughts and other drought types. The authors mention that the correlations are being calculated only for drought periods. Does that mean correlations are being calculated only for periods with SPI<-1 or when SSMI/SSFI/SGI<-1 or both? Please clarify.

**Response:** Thank you for pointing this out. We updated the text and clarify now that the lagged cross-correlations were computed using time periods where SSMI/SSFI/SGI were below -1. This ensures we are analyzing propagation time only to actual drought events (line 218)

R2_3: There are several interesting results in the study which have not been discussed in much detail. (1) The first of them being the high streamflow and groundwater drought propagation probabilities in cluster 4. While both cluster 4 and 5 are rainfall dominated, why are the probabilities so high for streamflow droughts in cluster 4 (Figure 6)? In L518-525, the authors have listed some factors which "could be" responsible for these differences such as soil water holding capacities but without much analysis. Soil type and land use land cover maps of the study region, if available, can be used to investigate the reasons for this observation. Also, why is there a significant difference between propagation probabilities of cluster 1 and 2 for streamflow and groundwater droughts in Figure 6?

**Response:** We appreciate this insightful comment. The aim of the correlation analysis presented in Section 3.4 is precisely to identify catchment characteristics that help explain differences in drought propagation probabilities and lag times across catchments/clusters. While we did consider variables such as soil type and land use, the analysis did not reveal any strong or consistent relationships between these factors and the propagation of streamflow or groundwater droughts. Instead, the analysis clearly pointed to total annual precipitation and streamflow as the dominant drivers of drought propagation characteristics. In general, wetter catchments tend to exhibit shorter propagation times and higher propagation probabilities - likely due to more dynamic hydrological responses and limited buffering capacity. This finding helps explain the patterns highlighted by the reviewer: Although both clusters 4 and 5 are rainfall-dominated, the most prominent and consistent difference between them (illustrated in Figure 2) is climatic. Cluster 4 receives approximately 35% more precipitation and generates nearly twice as much runoff as cluster 5. Similarly, cluster 1 receives about 50% more precipitation than cluster 2 and produces 150% more runoff. These contrasts align well with the results of the correlation analysis and provide a robust explanation for the substantially higher streamflow and groundwater drought propagation probabilities observed in cluster 4 compared to cluster 5, and in cluster 1 compared to cluster 2. We have now clarified this interpretation and strengthened the discussion accordingly (lines 543-548).

(2) Figure 7: While it is clear that the propagation probabilities for soil moisture droughts are highest in summer due to higher evaporative demand, the reasons for streamflow and groundwater drought propagation probabilities is not very clear. Why is propagation probability higher for streamflow and lower for groundwater in Autumn?

**Response:** Thank you for this observation. We would like to clarify that the differences in propagation probabilities for streamflow and groundwater across seasons, particularly the slightly higher streamflow and lower groundwater propagation probabilities in autumn, are not statistically significant, which is the reason why we have not further discussed this observation. We have clarified this in the results section (lines 361-366).

(3) Figure 8: While soil moisture propagation probabilities increase from north to south, the streamflow propagation probabilities decrease from north to south during summer season. Why?

The authors can explore the physical mechanisms underlying these interesting patterns observed in the statistical analysis. The authors may add maps of physical features such as soil type, vegetation, land use, runoff coefficients, baseflow index, snow fraction and geological features, if available, to better explain these patterns. Such analysis may reveal unique insights regarding drought propagation in high latitude watersheds.

**Response:** Thank you for highlighting this pattern. We had already tried to explain these contrasting patterns, but agree that we could be clearer. In our initial analysis, we deliberately chose not to include detailed spatial mapping of physical catchment features (which would add many more figures) and instead focused on the correlation analysis presented in Section 3.4 to detect overarching patterns and identify potential explanatory factors. However, we acknowledge that one key variable - the runoff coefficient - was discussed qualitatively but not included in the quantitative correlation analysis. To address this, we have taken the following steps: (1) we now report the average runoff coefficient for each cluster in the catchment description (Section 2.1, lines 131-133), (2) we have added the runoff coefficient as an explanatory variable in the correlation analysis to assess its role in shaping drought propagation dynamics (Section 3.5, lines 395 & 407), and (3) we have revised the discussion to reflect these additions (lines 580-582).

R2_4:   In Figure 9, the authors use correlation analysis to understand the physical factors affecting propagation probabilities and lag times. Physical catchment features like soil type and land use are also considered. However, I wonder if this analysis makes sense for soil moisture droughts considering that soil moisture droughts have been analyzed using reanalysis dataset in this study. Reanalysis models do not have a very accurate representation of soil types and land use features. Thus, conclusions based on this analysis could be misleading.

**Response:** This is an important point. We acknowledge that using reanalysis-based soil moisture may limit the validity of linking soil moisture droughts directly to physical catchment features like soil type or land use. We have now clarified this limitation in the discussion (lines 687-700).

Minor comments:

R2_5:   L494: I could not follow why places with delayed groundwater response would require more proactive management. Shouldn't it be reverse?

**Response:** Thank you, we agree this could be confusing. The key idea is that delayed groundwater response does not reduce the need for management, but actually increases the need for proactive management, because impacts emerge slowly but last longer. We have rephrased the sentence for clarification (lines 508-510).

R2_6:   L515: Please change to "other factors also".

**Response:** Changed (line 537).

R2_7:   L556: Please delete the extra comma.

**Response:** Done.

R2_8:   L600: How can annual rainfall and streamflow be used for drought forecasting? Please elaborate.

**Response:** Thank you for this important comment. We acknowledge that our original phrasing was not optimal. We agree that annual rainfall and streamflow alone cannot be used for direct drought forecasting. However, based on our analysis, these variables can support decision-making once drought conditions begin to emerge, specifically by helping to identify and prioritize areas that respond more quickly to changing hydrological conditions. In this context, annual rainfall and streamflow can serve as supplementary indicators in drought monitoring, guiding targeted resource allocation and mitigation efforts in faster-responding regions. We have rephrased this sentence (lines 625-628).

**References**

Bloomfield, J.P., Marchant, B.P., 2013. Analysis of groundwater drought building on the standardised precipitation index approach. Hydrol. Earth Syst. Sci. 17, 4769–4787. https://doi.org/10/f5m79s

Geng, G., Zhang, B., Gu, Q., He, Z., Zheng, R., 2024. Drought propagation characteristics across China: Time, probability, and threshold. J. Hydrol. 631, 130805. https://doi.org/10.1016/j.jhydrol.2024.130805

Gong, R., Chen, J., Liang, Z., Wu, C., Tian, D., Wu, J., Li, S., Zeng, G., 2023. Characterization and propagation from meteorological to groundwater drought in different aquifers with multiple timescales. J. Hydrol. Reg. Stud. 45, 101317. https://doi.org/10.1016/j.ejrh.2023.101317

Hellwig, J., Liu, Y., Stahl, K., Hartmann, A., 2022. Drought propagation in space and time: the role of groundwater flows. Environ. Res. Lett. 17, 094008. https://doi.org/10.1088/1748-9326/ac8693

Ma, L., Huang, Q., Huang, S., Liu, D., Leng, G., Wang, L., Li, P., 2021. Propagation dynamics and causes of hydrological drought in response to meteorological drought at seasonal timescales. Hydrol. Res. 53, 193–205. https://doi.org/10.2166/nh.2021.006

Odongo, R.A., De Moel, H., Van Loon, A.F., 2023. Propagation from meteorological to hydrological drought in the Horn of Africa using both standardized and threshold-based indices. Nat. Hazards Earth Syst. Sci. 23, 2365–2386. https://doi.org/10.5194/nhess-23-2365-2023

Sattar, M.N., Lee, J.-Y., Shin, J.-Y., Kim, T.-W., 2019. Probabilistic Characteristics of Drought Propagation from Meteorological to Hydrological Drought in South Korea. Water Resour. Manag. 33, 2439–2452. https://doi.org/10.1007/s11269-019-02278-9

Wang, H., Zhu, Y., Qin, T., Zhang, X., 2022. Study on the propagation probability characteristics and prediction model of meteorological drought to hydrological drought in basin based on copula function. Front. Earth Sci. 10. https://doi.org/10.3389/feart.2022.961871

Xu, Z., Wu, Z., Shao, Q., He, H., Guo, X., 2023. From meteorological to agricultural drought: Propagation time and probabilistic linkages. J. Hydrol. Reg. Stud. 46, 101329. https://doi.org/10.1016/j.ejrh.2023.101329

Zhang, Q., Miao, C., Gou, J., Wu, J., Jiao, W., Song, Y., Xu, D., 2022. Spatiotemporal characteristics of meteorological to hydrological drought propagation under natural conditions in China. Weather Clim. Extrem. 38, 100505. https://doi.org/10.1016/j.wace.2022.100505